# Dual-path Collaborative Generation Network for Emotional Video Captioning

Cheng Ye
kyrieye@mail.ustc.edu.cn
School of Information Science and
Technology, USTC
Hefei, China

Weidong Chen*
chenweidong@ustc.edu.cn
School of Information Science and
Technology, USTC
Hefei, China

Jingyu Li
jingyuli@mail.ustc.edu.cn
School of Information Science and
Technology, USTC
Hefei, China

Lei Zhang
leizh23@ustc.edu.cn
School of Information Science and
Technology, USTC
Hefei, China

Zhendong Mao
zdmao@ustc.edu.cn
School of Information Science and
Technology, USTC
Hefei, China

## ABSTRACT

Emotional Video Captioning (EVC) is an emerging task that aims to describe factual content with the intrinsic emotions expressed in videos. The essential of the EVC task is to effectively perceive subtle and ambiguous visual emotional cues during the caption generation, which is neglected by the traditional video captioning. Existing emotional video captioning methods perceive global visual emotional cues at first, and then combine them with the video features to guide the emotional caption generation, which neglects two characteristics of the EVC task. Firstly, their methods neglect the dynamic subtle changes in the intrinsic emotions of the video, which makes it difficult to meet the needs of common scenes with diverse and changeable emotions. Secondly, as their methods incorporate emotional cues into each step, the guidance role of emotion is overemphasized, which makes factual content more or less ignored during generation. To this end, we propose a dual-path collaborative generation network, which dynamically perceives visual emotional cues evolutions while generating emotional captions by collaborative learning. The two paths promote each other and significantly improve the generation performance. Specifically, in the dynamic emotion perception path, we propose a dynamic emotion evolution module, which first aggregates visual features and historical caption features to summarize the global visual emotional cues, and then dynamically selects emotional cues required to be re-composed at each stage as well as re-composed them to achieve emotion evolution by dynamically enhancing or suppressing different granularity subspace's semantics. Besides, in the adaptive caption generation path, to balance the description of factual content and emotional cues, we propose an emotion adaptive decoder, which firstly estimates emotion intensity via the alignment of emotional features and historical caption features at each generation step, and then, emotional guidance adaptively incorporate into the caption generation based on the emotional intensity. Thus, our methods can generate emotion-related words at the necessary time step, and our caption generation balances the guidance of factual content and emotional cues well. Extensive experiments on three challenging datasets demonstrate the superiority of our approach and each proposed module.[1]

## CCS CONCEPTS

• **Computing methodologies** → **Scene understanding**.

## KEYWORDS

Video Captioning, Emotion Learning, Collaborative Generation

**ACM Reference Format:**
Cheng Ye, Weidong Chen, Jingyu Li, Lei Zhang, and Zhendong Mao. 2024. Dual-path Collaborative Generation Network for Emotional Video Captioning. In *Proceedings of the 32nd ACM International Conference on Multimedia (MM '24), October 28-November 1, 2024, Melbourne, VIC, AustraliaProceedings of the 32nd ACM International Conference on Multimedia (MM'24), October 28-November 1, 2024, Melbourne, Australia.* ACM, New York, NY, USA, 10 pages. https://doi.org/10.1145/3664647.3681603

## 1 INTRODUCTION

The widespread popularity of self-media platforms, such as TikTok, Twitter and Facebook, allows everyone to express their opinions and emotions through the created video content. Meanwhile, the created videos are easy to arouse a wide range of emotional responses from viewers due to their vividness and variability. Under such circumstances, in the face of the increasing number of videos online, emotion-based video understanding tasks have attracted more and more attention, including video emotional recognition[5, 14, 22, 47], video advertising recommendation [7, 9, 18, 22, 28], and emotional video captioning (EVC) [1, 34, 35, 41]. The EVC task is an emerging hot topic in multimedia (vision-and-language) communities, which requires not only understanding the factual video contents, but also

*Corresponding Author.

[1]Our code is publicly available at https://github.com/kyrieye/MM-2024.

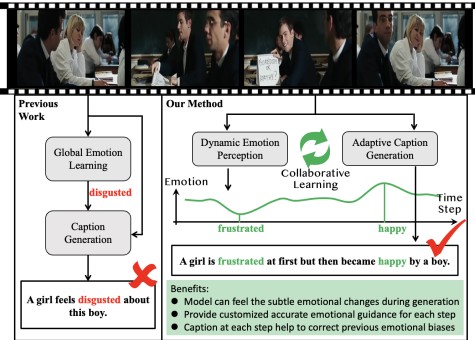

**Figure 1: Motivation of our method, which collaborates the dynamic emotion perception and adaptive caption generation. Our method can generate accurate emotional captions.**

recognizing the complex emotion cues contained in the video and incorporating the emotion semantics to generate captions.

The EVC task is the extension of the traditional video captioning task [10, 42, 50, 51]. Most existing video captioning methods focus on mining factual semantics, such as identifying visual objects and their attributes, associating their relationships, and translating them into text descriptions. However, they neglect the intrinsic emotions conveyed in the video contents, which makes the generated sentences a bit boring and soulless [1]. In contrast, merely modeling factual semantics in EVC task is insufficient to fill the affective gap between videos and emotions. For example, for the caption "the little girl loses one baby teeth but feels a bit of delightful". The vision factual semantic for "lost teeth" and the emotional semantic "delightful" lead to a gap. Thus, the essential of the EVC task is to effectively perceive subtle and ambiguous visual emotional cues during the caption generation.

To fill the affective gap between videos and emotions, existing emotional video captioning methods make their effort to perceive emotion by global emotion learning in the initial stage, and then combine it with video features to generate emotional descriptions. Despite the promising progress, they neglect two characteristics of the EVC task. Firstly, their methods neglect the dynamic subtle changes in the video's intrinsic emotions, making it difficult to meet the needs of complex scenes with various emotions. For example, as shown in Figure 1, the emotion of the girl in the video changes from frustrated to happy. Existing methods overlook capturing these subtle emotional changes at different time periods of the video and fail to generate accurate multi-emotional related descriptions. Secondly, as they incorporate emotional cues into each generation step, the guidance role of emotion is overemphasized, which will make factual content more or less ignored when generating descriptions. Moreover, when previous EVC methods predict the wrong global emotions, their methods inevitably generate descriptions that are not closely related to the factual contents and have nothing to do with the correct visual emotion cues.

To fill the research gap, we propose a **D**ual-path **C**ollaborative **G**eneration **N**etwork to adaptively generate emotional captions word-by-wordly while dynamically perceiving subtle emotion evolutions. Our dynamic emotion perception is not independent of

the caption generation. At each generation step, in the dynamic emotion perception path, the subtle emotional evolution from the previous step is perceived according to the combination of historical caption features and visual features; whereas, in the adaptive caption generation path, our method adaptively generates emotional captions based on the evolved emotions. The proposed two paths promote each other by our collaborative learning, leading to a promising performance in emotional captions generation.

In the dynamic emotion perception path, our method perceives subtle emotional evolutions for each generation step. Thus, on the one hand, our dynamic emotion perception can correct the emotion bias of previous steps, helping to generate the correct emotion related words at the necessary step; whereas, on the other hand, our method can adapt to real-world scenarios with diverse and changeable emotions, providing the most accurate emotional guidance for each generation step. Specifically, we propose a dynamic emotion evolution module, which first aggregates visual features and historical caption features to summarize the global visual emotional cues, and then dynamically selects emotional cues required to be re-composed at each stage as well as re-compose them to achieve emotion evolution by dynamically enhancing or suppressing different granularity sub-space's semantic.

In the adaptive caption generation path, the captions generated by our method are not biased towards emotional cues or factual content. Instead, our methods perceive the correct emotions and adaptively generate emotion-related words at the necessary generation steps, making our captions vivid and diverse. To be specific, an elegant emotion-adaptive decoder is designed, which firstly leverages the alignment of the perceived evolved emotions and historical caption features to estimate the emotion intensity at the current generation step. If the perceived evolved emotions and historical captions are harmonious and unified, it means that emotion-related words need to be generated at this step, and vice versa. Then, our method adaptively integrates the perceived emotions into the caption generation with the guidance of the estimated emotion intensity. Extensive experiments on three challenging datasets demonstrate the superiority of our approach and each proposed module in both quantitative and qualitative terms.

In short, our main contributions are summarized as follows:

• We proposed a novel **D**ual-path **C**ollaborative **G**eneration **N**etwork for emotional video captioning, which dynamically perceives emotion cue evolution while generating emotional captions. The proposed two paths promote each other by collaborative learning, facilitating the emotional captions generation performance.

• In the dynamic emotion perception path, a dynamic emotion evolution module is designed, which dynamically recomposes emotion features to achieve emotion evolution at different generation steps, helping to correct the previous emotion bias and providing customized and accurate emotion guidance for caption generation.

• In the adaptive caption generation path, we propose an emotion adaptive decoder that utilizes the alignment of evolved emotion and historical caption to evaluate the emotional intensity for the current state, which helps our method generate emotion-related words at the necessary time step, and balances the guidance of factual content and emotional cues well.

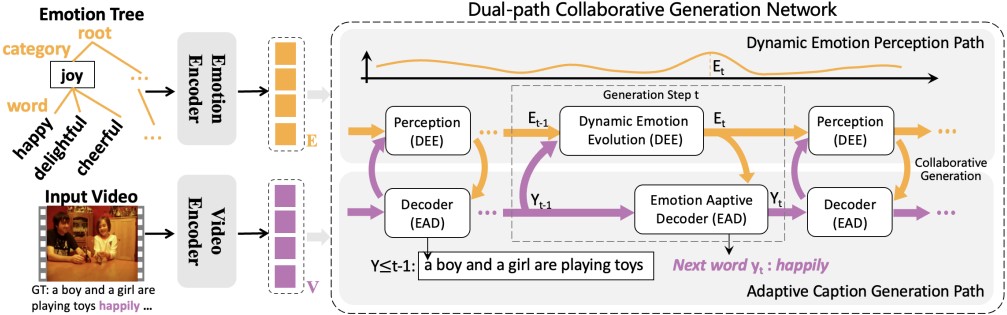

**Figure 2: The overview of our proposed dual-path collaborative generation network. It mainly consists of the dynamic emotion perception and the adaptive caption generation.**

• Experiments on three public datasets (*e.g.*, EVC-MSVD, EVC-VE, and EVC-Combined) demonstrate the superiority of our framework, *e.g.*, improving the latest records by +7.2%, +6.8% and 6.5% w.r.t. emotion accuracy, CIDEr and CFS, respectively, on EVC-VE.

## 2 RELATED WORK

### 2.1 Visual Emotional Analysis

In recent years, instead of only focusing on factual semantic, many researchers have attempted to understand the emotional cues contained in visual elements[6, 19, 37, 44, 57]. Existing visual emotional analysis can be divided into several categories: 1) emotion classification task based on categorical emotion states from the psychological view[29, 48, 54]. Yang et al. [52] explore the relationship between scenes and objects to build emotional graphs, and leverage GCN for reasoning to achieve emotional representation. Zhang et al. [55] integrate hierarchical information of images to generate high-quality emotional representation. 2) emotion recognition from human activity *e.g.*, facial expression and action[8, 13, 15, 23, 25]. Wang et al. [43] utilize a self-attention mechanism over mini-batch and a careful relabeling mechanism to suppress uncertainties for facial expression recognition. Zhao et al. [58] propose a label distribution learning method as a novel training strategy for robust facial expression recognition. 3) emotion distribution learning from complex emotion analysis[20, 36, 49]. Yang et al. [53] propose a circular-structured representation based on emotion circles to promote emotion distribution learning. Jia et al. [20] employ a local low-rank structure to capture the local label correlations implicitly. In video-based emotion analysis, Mittal et al. [31] explicitly model the temporal causality using attention-based methods and Granger causality for time-series emotion prediction for multimedia content. Zhao et al. [56] leverage a hierarchical attention mechanism to achieve end-to-end video emotion recognition. Motivated by these works, we integrate emotional cues mining into the video captioning task to generate vivid descriptions.

### 2.2 Emotional Video Captioning

Early works usually leverage pre-defined emotion categories (*e.g.*, anger, disgust, fear, happiness, sadness, or surprise) to output emotionally customized captions, while ignoring the emotional cues

elicited by visual content. To break this, the emotional video captioning task has attracted interest in the community, which firstly performs emotional analysis on the visual content, and then inputs the emotional representation into the decoder to generate emotional descriptions. Wang et al. [41] release a new dataset of emotional video captioning. They design two independent prediction networks for fact flow and emotion flow, then use a weighted average for the output probabilities of the two modules to generate descriptions. Song et al. [34] propose a contextual attention network to recognize and describe the fact and emotion in the video by semantic-rich context learning. Song et al. [1] propose a novel tree-structured emotion learning module to achieve explicit emotion perception. Song et al. [35] incorporate visual context, textual context, and visual-textual relevance into an aggregated multimodal contextual vector to enhance video captioning. However, existing works ignore the fact that emotions vary slightly across a video stream. Thus, in this work, we study dynamically changing emotion representations to generate fine-grained emotional captions.

## 3 PROPOSED METHOD

The framework of our proposed method is illustrated in Figure 2, which mainly consists of three parts: 1) Video and Emotion Feature Extraction, 2) Dynamic Emotion Perception Path and 3) Adaptive Caption Generation Path.

### 3.1 Video and Emotion Feature Extraction

**Video Encoder.** The EVC task aims to generate emotion-dominated descriptions of input video streams. Following the setting of previous work [1, 34], we firstly down-sample the given video to get a set of frames, and then, we leverage 2-D CNN (*e.g.*, ResNet [17] pre-trained on ImageNet [11]) and 3-D CNN (*e.g.*, 3D-ResNeXt [16] pre-trained on Kinetics [3]) to extract appearance features and motion features, respectively. Next, we utilize the Transformer block to further encode appearance features and motion features to obtain the final video features $V \in \mathbb{R}^{N \times d_v}$, which can be formalized as:

$$F = [\text{Linear}(F_a); \text{Linear}(F_m)], \tag{1}$$

$$V = \text{Transformer}_v(F)|_{\{Q,K,V\}:F}, \tag{2}$$

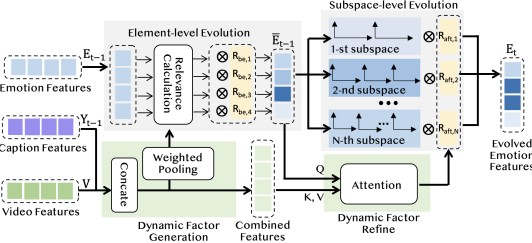

**Figure 3: The framework of our dynamic emotion evolution.**

where $[;]$ is the concatenate operation, $F_a$ and $F_m$ are appearance features and motion features, respectively. The obtained video features $V$ capture factual semantics, *e.g.*, objects and scenes, etc.

**Emotion Encoder.** Following [1], we leverage the tree-structured emotion learning to generate hierarchical emotion representations. Firstly, we obtain the video emotion category representations. Given video feature $V$ and emotion category dictionary $D_c = \{c_i\}_{i=1}^{N_c}$, where $c_i$ denotes the $i$-th category such as "love" and $N_c$ denotes the number of emotion category. The emotion category representations $F_c$ can be encoded by a Transformer block as:

$$F_c = \text{Transformer}_c(D_c, V)|_{Q:V, \{K,V\}:D_c}, \qquad (3)$$

and the emotion category distribution $P_c$ over $D_c$ is predicted based on $F_c$ by average pooling and linear mapping operation. We learn fine-grained emotion word representations based on $P_c$ by masked attention mechanism. Given $P_c$ and emotion word dictionary $D_w = \{w_i\}_{i=1}^{N_w}$, we only consider the top-K emotional categories, and select the emotional words contained in these K categories to generate the emotion mask 0-1 matrix $A \in \mathbb{R}^{N \times N_w}$. Then, we impose the mask $A$ onto $V$ and $D_w$ through a transformer block to obtain the emotion word representation $E \in \mathbb{R}^{N \times d_e}$:

$$E = \text{Transformer}_w(D_w, V, mask = A)|_{Q:V, \{K,V\}:D_w}, \qquad (4)$$

where $E$ is the global emotional cues based on video features and hierarchical emotion tree structure, which also is the initial state $E_0$ of our dynamic emotional perception path. Then, the emotion word distribution $P_w$ over $D_w$ is predicted based on $E$.

## 3.2 Dynamic Emotion Perception Path

In the dynamic emotion perception path, our method can adapt to complex scenes with diverse and changeable emotions, and in each generation step, it can perceive subtle emotional evolution and provide the most accurate emotional guidance for generation. Specifically, given video features $V \in \mathbb{R}^{N \times d_v}$, the caption features of the historically generated captions $Y^{t-1} \in \mathbb{R}^{t-1 \times d_t}$, and current emotion features $E^{t-1} \in \mathbb{R}^{N \times d_e}$, our module finds $E^{t-1}$'s diverse semantic information by element-level and subspace-level, and finally aggregates this information to reorganize features to obtain evolved emotion representations $E^t \in \mathbb{R}^{N \times d_e}$.

**1) Dynamic Factor Generation.** In the process of generating captions for videos, historical caption features represent the current generation state, while video content features reveal subtle emotional changes. We employ a weighted pooling approach on the caption and video combined features to effectively capture these

changes in the current generation state. Specifically, we first project the video features into the same embedding space as the caption features and combine them to obtain the combined features $C$:

$$C = [W_v V; Y^{t-1}] \in \mathbb{R}^{(2(t-1)) \times d_t}, \qquad (5)$$

where $W_v$ and $[;]$ are learnable parameters and the concatenate operation, respectively. Then, we leverage the weighted pooling to reconstruct the combined features into the same embedding space with the emotion feature to obtain the extended feature $\overline{Y}^{t-1}$:

$$\overline{Y}^{t-1} = (\text{Softmax}(C \cdot W_c^1))^T (C \cdot W_c^2) \in \mathbb{R}^{M \times d_e}, \qquad (6)$$

where $W_c^1 \in \mathbb{R}^{d_t \times M}$ and $W_c^2 \in \mathbb{R}^{d_t \times d_e}$ are learnable parameters and $M$ is a hyper-parameter that controls the size of the feature space. Compared to the original feature, the extended features not only include the simple caption status, but also consider the changes in the emotion of the video content, which provides accurate support for element-level emotional evolution.

**2) Element-level Emotion Evolution.** After obtaining the extended features, we construct the element-level emotion evolution by selecting the emotion sub-features that need to be reorganized. Human emotions are complex and changeable, and different caption generation stages should have different emotional focuses. Thus, it is necessary to eliminate the negative impact of irrelevant emotional elements. In order to select irrelevant emotional elements, we firstly calculate the cross-modality relevance to align the current emotion features $E^{t-1}$ and the extended features $\overline{Y}^{t-1}$. Then, we concatenate the aligned extended features and emotion features, and project them to an N-vector to obtain the dynamic emotion factors $R_{be}$. Finally, we leverage $R_{be}$ to get element-level evolved emotion feature $\overline{E}^{t-1}$ which required to be re-composed next:

$$A' = \text{Softmax}(\text{Pooling}(E^{t-1}(\overline{Y}^{t-1})^T)) \in \mathbb{R}^{1 \times M}. \qquad (7)$$

$$R_{be} = \text{Relu}([E^{t-1}; (A'\overline{Y}^{t-1})]W_R) \in \mathbb{R}^{1 \times N}, \qquad (8)$$

$$\overline{E}^{t-1} = R_{be} \odot E^{t-1} \in \mathbb{R}^{N \times d_e}, \qquad (9)$$

where $W_R \in \mathbb{R}^{d_e \times N}$ is a learnable weight, and $\odot$ is the element-wise multiplication. By selectively reorganizing the emotion elements based on their relevance to the dynamic emotion factors, the model can focus on the most relevant emotional aspects at each time step.

**3) Subspace-level Emotion Re-composition.** The feature space, such as $\mathbb{R}^{d_e}$, can be divided into different subspace representations, e.g., $\mathbb{R}^{k \times \frac{d_e}{k}}$, which contain different semantic information, helping us capture subtle emotional changes. Thus, given the element-level evolved emotion features $\overline{E}^{t-1}$ and a subspace number list $\{k_j | j = 1, 2, .., N_k\}$, where $N_k$ denotes the number of subspace, we divide $\overline{E}^{t-1}$ into $\overline{E}_j^{t-1}$ for each $k_j$ as:

$$\overline{E}^{t-1} \in \mathbb{R}^{N \times d_e}, \overline{E}_j^{t-1} \in \mathbb{R}^{N \times k_j \times \frac{d_e}{k_j}}. \qquad (10)$$

Then, the semantic information of each subspace is captured by the attention mechanisms, which can be formalized as:

$$O_j = \text{Tanh}(\text{Atten}_e(\overline{E}^{t-1}, \overline{Y}^{t-1})|_{Q:\overline{E}^{t-1}, \{K,V\}:\overline{Y}^{t-1}}) \in \mathbb{R}^{N \times k_j}, \quad (11)$$

$$E_j^t = \text{Reshape}(\overline{E}_j^{t-1} \odot O_j) \in \mathbb{R}^{N \times d_e}, \qquad (12)$$

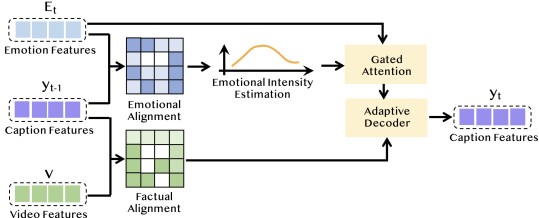

**Figure 4: The framework of our adaptive caption generation.**

where $\odot$ is the element-wise multiplication. Reshape operation turns the emotion features back to the original size $R^{N \times d_e}$. Thus, the semantics of different subspaces are reorganized to varying degrees. To merge the semantic information of different subspaces to generate new emotional features, we leverage another attention to generate the weights for each subspace $R_{aft} \in \mathbb{R}^{N_k}$:

$$R'_j = \text{Atten}_w(\overline{E}^{t-1}, \overline{Y}^{t-1})|_{Q:\overline{E}^{t-1}, \{K,V\}:\overline{Y}^{t-1}} \in \mathbb{R}^{N \times 1}, \quad (13)$$

$$R_{aft} = \text{Softmax}(\text{Pooling}(R')) \in \mathbb{R}^{N_k} \quad (14)$$

Finally, we apply the weighted value to each reorganized subspace features to generate evolved emotional features:

$$E^t = E^{t-1} + \sum_{j=1}^{N_k} (R_{aft,j} E^t_j). \quad (15)$$

where $R_{aft,j}$ is the $j$-th element of the $R_{aft}$.

By capturing emotional changes at each generation step, our Dynamic Emotion Perception Path ensures that the generated captions are emotionally relevant and synchronized with the video content. Moreover, the multi-step nature of the emotion evolution modules allows our methods to the fine-grained analysis of emotional dynamics, enabling the generation of captions that accurately reflect the emotional nuances of the video content.

## 3.3 Adaptive Caption Generation Path

There are different demands of emotional cues for different stages in emotional caption generation. To this end, in our adaptive caption generation path, we propose the emotion adaptive decoder (EAD) which adaptively adjusts the guidance intensity of emotional semantics when generating captions to avoid the loss of factual semantics caused by overemphasis on emotional semantics.

**1) Emotion Intensity Estimation.** The historical caption features help determine whether emotional guidance is needed for the next generation step. For example, when generating adjectives or adverbs (*e.g.*, surprised and angrily) in sentences, emotional features have strong guiding significance. Instead, when generating nouns (*e.g.*, bike and hospital), factual semantics in video features have stronger guiding significance than emotional features. Thus we propose the emotion intensity estimation module that leverages caption features to estimate emotion guidance intensity at the current generation step. Mathematically, at the $t$-th time-step, given caption features $Y^{t-1} \in \mathbb{R}^{(t-1) \times d_t}$ and evolved emotion features $E^t \in \mathbb{R}^{N \times d_e}$, we firstly obtain emotion correlated text features $\mathcal{T}_e^t$ by calculating the normalized relevance score to align each generated word $y_j^{t-1}$ onto each emotion element $e_j^t$. Then, we apply the

gating unit to $\mathcal{T}_e^t$ to obtain the emotion guidance intensity $g^t$.

$$r_{i,j}^t = \text{Tanh}(W_r y_i^{t-1} + H_r e_j^t + b_r), \quad (16)$$

$$\mathcal{T}_{e,i}^t = \sum_{j=1}^{t} \text{Softmax}(r_{i,j}^t) y_j^{t-1}, i \in [1, N]. \quad (17)$$

$$g^t = \text{Sigmoid}(W_g \mathcal{T}_e^t + b_g) \in \mathbb{R}^N, \quad (18)$$

where $y_i^{t-1}$ represents the $i$-th features of the caption features $Y^{t-1}$, and $e_j^t$ is the $j$-th features of emotion features $E^t$.

**2) Adaptive Caption Generation.** Since previously generated words also help predict the next word, we obtain the visually correlated text feature $\mathcal{T}_v^t$ like $\mathcal{T}_e^t$. At the decoding step t, we aggregate all the sufficient semantics and use emotion guidance intensity to adapt the features. Finally, we obtain the probability distribution $P(y_t) \in \mathbb{R}^D$ of the next word $y^t$, which is defined as:

$$X^t = [V, \mathcal{T}_v^t, g^t \odot E^t] \in \mathbb{R}^{N \times (d_v + d_t + d_e)} \quad (19)$$

$$h^t = \text{Decoder}(\text{Pooling}(X^t), h^{t-1}), \quad (20)$$

$$P(y_t) = W_p h^t + b_p. \quad (21)$$

where Decoder is a standard attention-based LSTM to generate the caption sentence step by step, $W_p$ and $b_p$ are learnable parameters.

To ensure both the factual and emotional semantic correctness of the generated captions, we leverage two objective functions to guide the correct generation direction. Due to the importance of emotional words in captions, we leverage emotion-focused cross-entropy loss $\mathcal{L}_e$ that adds a penalty term on emotional words, which ensures the correctness of generated emotional words:

$$\mathcal{L}_e = \begin{cases} -(1 + \beta) \sum_t \log P(y_t), & if : y_t \in D_w, \\ -\sum_t \log P(y_t), & otherwise, \end{cases} \quad (22)$$

where $\beta$ is a hyper-parameter that control the level of punishment while $y_t$ is a emotional word, like "happily".

Additionally, the initial distribution of emotions is critical to the direction of emotion evolution. Thus we calculate the hierarchical initial emotional classification loss $\mathcal{L}_{cls}$ based on $P_c$ and $P_w$:

$$\mathcal{L}_{cls} = -\sum_{x \in \overline{Y}_c} \log P_c(x) - \sum_{x \in \overline{Y}_w} \log P_w(x), \quad (23)$$

where $\overline{Y}_c$ and $\overline{Y}_w$ denote the ground-truth emotion category and word, respectively. Finally, our model can be trained by minimizing the weighted sum of these two losses:

$$\mathcal{L} = \lambda_e \mathcal{L}_e + \lambda_{cls} \mathcal{L}_{cls}, \quad (24)$$

where $\lambda_e$ and $\lambda_{cls}$ are two hyper-parameters that aim to control the balance of two losses.

# 4 EXPERIMENTS

## 4.1 Datasets and Evaluation Metrics

**Dataset.** We experiment on three public emotional video captioning benchmarks, *e.g.*, EmVidCap-S[4], EmVidCap-L[21], and EmVidCap[41]. **EmVidCap-S** is built via additionally embedding emotion words into the caption sentences of traditional video captioning dataset MSVD[4], which contains 240/134 videos and

**Table 1: Comparison with the state-of-the-art methods on three challenging datasets, *i.e.*, EVC-MSVD, EVC-VE, and EVC-Combine. Our method outperforms the state-of-the-art methods for all metrics. The best results are highlighted in bold.**

| Dataset | Methods | Features | $\text{Acc}_{sw}$ | $\text{Acc}_c$ | BLEU-1 | BLEU-2 | BLEU-3 | BLEU-4 | METEOR | ROUGE | CIDEr | BFS | CFS |
|---|---|---|---|---|---|---|---|---|---|---|---|---|---|
| EVC-MSVD | FT[41] | R152 | 69.4 | 67.1 | 77.2 | 60.3 | 47.4 | 36.3 | 29.0 | 63.4 | 62.5 | 52.5 | 63.7 |
| | CANet[34] | | 69.6 | 66.4 | 78.0 | 62.1 | 50.0 | 38.6 | 29.8 | 63.5 | 63.3 | 54.1 | 64.2 |
| | VEIN[35] | | 72.3 | 69.3 | 78.6 | 61.6 | 48.4 | 37.0 | 29.4 | 63.2 | 64.7 | 53.8 | 65.9 |
| | EPAN[1] | | 79.0 | 76.1 | 80.4 | 64.7 | 53.0 | 43.1 | 32.0 | 66.4 | 69.8 | 58.8 | 71.4 |
| | **DCGN(Ours)** | | **81.2** | **80.5** | **81.7** | **66.9** | **55.4** | **45.3** | **33.1** | **69.7** | **70.8** | **60.1** | **73.0** |
| | CANet[34] | R101+RN | 78.7 | 76.8 | 78.5 | 64.0 | 52.1 | 41.8 | 30.8 | 65.7 | 74.4 | 57.9 | 75.1 |
| | VEIN[35] | | 80.1 | 79.1 | 79.6 | 64.4 | 52.9 | 42.7 | 31.7 | 66.9 | 71.1 | 59.0 | 72.8 |
| | EPAN[1] | | 84.4 | 82.8 | 79.8 | 65.8 | 53.5 | 41.9 | 33.0 | 66.8 | 75.7 | 59.9 | 77.3 |
| | **DCGN(Ours)** | | **86.7** | **85.7** | **82.0** | **66.7** | **56.0** | **45.1** | **33.9** | **70.0** | **76.7** | **62.0** | **79.4** |
| | SA-LSTM[39] | CLIP | 68.8 | 67.2 | 80.7 | 67.9 | 56.3 | 45.5 | 33.0 | 68.2 | 72.1 | 59.0 | 71.3 |
| | VEIN[35] | | 82.7 | 82.1 | 82.0 | 68.4 | 57.1 | 45.9 | 33.0 | 69.0 | 79.6 | 62.4 | 80.2 |
| | EPAN[1] | | 84.1 | 82.8 | 82.5 | 69.6 | 57.8 | 46.2 | 34.4 | 69.8 | 80.6 | 63.1 | 81.1 |
| | **DCGN(Ours)** | | **86.5** | **85.7** | **84.5** | **70.9** | **59.2** | **48.7** | **35.7** | **71.0** | **85.2** | **65.7** | **86.6** |
| EVC-VE | CANet[34] | R152 | 42.5 | 40.2 | 66.0 | 44.6 | 29.1 | 18.8 | 17.7 | 37.7 | 23.9 | 33.7 | 27.4 |
| | EPAN[1] | | 47.5 | 45.8 | 68.4 | 46.8 | 31.2 | 20.8 | 18.7 | 38.9 | 26.4 | 36.5 | 30.4 |
| | **DCGN(Ours)** | | **49.8** | **48.1** | **69.9** | **48.0** | **33.5** | **22.3** | **19.8** | **40.5** | **28.7** | **38.9** | **33.4** |
| | CANet[34] | R101+RN | 41.9 | 39.7 | 66.9 | 44.8 | 29.3 | 19.3 | 18.2 | 37.9 | 23.3 | 33.9 | 26.8 |
| | EPAN[1] | | 49.3 | 47.9 | 67.7 | 45.9 | 30.9 | 21.1 | 18.5 | 38.3 | 24.8 | 36.6 | 29.5 |
| | **DCGN(Ours)** | | **51.3** | **50.6** | **69.6** | **48.2** | **33.8** | **22.1** | **19.5** | **42.0** | **28.4** | **38.7** | **33.1** |
| | SA-LSTM[39] | CLIP | 48.6 | 47.1 | 71.0 | 51.1 | 34.5 | 22.5 | 19.6 | 40.7 | 30.2 | 38.9 | 33.7 |
| | EPAN[1] | | 63.8 | 62.3 | 73.6 | 54.0 | 38.3 | 27.0 | 21.2 | 42.3 | 34.7 | 45.0 | 40.4 |
| | **DCGN(Ours)** | | **71.0** | **69.4** | **74.5** | **55.3** | **40.0** | **28.1** | **23.4** | **47.7** | **41.5** | **47.3** | **46.9** |
| EVC-Combine | FT[41] | R152 | 51.2 | 49.6 | 67.6 | 47.2 | 32.0 | 21.6 | 20.4 | 43.1 | 29.0 | 37.6 | 33.3 |
| | VEIN[35] | | 52.2 | 50.6 | 67.3 | 46.9 | 32.3 | 21.9 | 19.6 | 43.2 | 30.3 | 37.9 | 34.5 |
| | EPAN[1] | | 53.3 | 51.5 | 68.0 | 47.4 | 32.1 | 21.5 | 19.7 | 42.8 | 31.3 | 38.1 | 35.5 |
| | **DCGN(Ours)** | | **55.4** | **53.0** | **69.7** | **49.1** | **33.9** | **22.6** | **21.7** | **44.3** | **33.7** | **40.2** | **37.9** |
| | CANet[34] | R101+RN | 53.7 | 52.7 | 68.1 | 47.7 | 32.9 | 22.5 | 19.7 | 43.7 | 34.5 | 38.8 | 38.2 |
| | VEIN[35] | | 58.1 | 57.0 | 69.7 | 49.4 | 35.3 | 24.5 | 20.7 | 45.7 | 37.1 | 41.4 | 41.2 |
| | EPAN[1] | | 60.3 | 59.7 | 70.7 | 50.7 | 35.4 | 24.5 | 20.8 | 46.0 | 36.9 | 42.1 | 41.6 |
| | **DCGN(Ours)** | | **63.4** | **63.1** | **71.4** | **52.0** | **38.2** | **26.7** | **22.4** | **48.7** | **39.5** | **45.1** | **44.3** |
| | SA-LSTM[39] | CLIP | 53.4 | 50.7 | 70.6 | 51.4 | 36.7 | 25.4 | 21.0 | 45.9 | 38.8 | 41.2 | 41.5 |
| | VEIN[35] | | 59.0 | 57.6 | 72.1 | 52.8 | 37.9 | 27.1 | 21.6 | 46.8 | 39.4 | 43.6 | 43.1 |
| | EPAN[1] | | 69.3 | 67.2 | 74.4 | 55.6 | 39.9 | 28.0 | 23.0 | 47.1 | 43.0 | 47.0 | 48.0 |
| | **DCGN(Ours)** | | **74.8** | **73.1** | **75.6** | **56.7** | **40.5** | **28.5** | **24.9** | **51.7** | **49.8** | **48.5** | **51.7** |

8,169/4,611 sentences for training/testing, respectively. **EmVidCap-L** is constructed by writing the caption sentences with emotion expression based on the emotion prediction dataset VideoEmotion-8 [21]. The EmVidCap-L is divided into 1,141/382 videos and 19,398/6,527 sentences for training/testing, respectively. **EmVidCap** is the combination of EmVidCap-S and EmVidCap-L, which contains 1,381/516 videos and 27,567/11,138 sentences for training/testing.

**Evaluation Metrics.** To evaluate the accuracy of emotion in generated sentences effectively, following previous work[41], we consider the emotion word accuracy $\text{Acc}_{sw}$ [41] and emotion sentence accuracy $\text{Acc}_c$ [41]. Additionally, to measure the semantics information of generated captions, we use the standard metrics*e.g.*, **BLEU**[32], **METEOR**[2], **ROUGE**[26], and **CIDEr**[38]. Moreover, there are two hybrid metrics **BFS**[41] and **CFS**[41] that combine the emotion evaluation with BLEU and CIDEr metrics, respectively.

## 4.2 Implementation Details

For a fair comparison, we extract appearance features by leveraging ResNet-101 [17] or ResNet-152 [17] and motion features using 3D-ResNext-101 [16]. Besides, we test the model with the modern CLIP features [33]. The maximum length of captions is set to 15. We build an overall vocabulary that contains all words in the corpus. The vocabulary sizes for the EVC-MSVD, EVC-VE, and EVC-Combined datasets are 9,637, 13,980, and 14,034, respectively. For emotion learning, there are $N_c$ = 34 emotion categories and $N_w$ = 179 emotion words. All word embeddings are extracted by GloVe. The feature dimension is set to $d_v = d_t = d_e = 300$. For dynamic emotion perception, the extension parameter is set to $M = 100$ and the subspace number list is set to $\{2, 3, 5, 6, 10\}$ to construct a complete semantic space since the dimension of emotion features is $d_e = 300$. The hidden size of the caption decoder is set to 512. We set the penalty parameter $\beta$, the objective function weight $\lambda_e$ and $\lambda_{cls}$ to 0.1, 1 and 0.2. We adopt the Adam optimizer with a learning rate of 7e-4, and the batch size is set to 128.

## 4.3 Main Results

**Comparison with State-of-the-Art Methods.** As shown in Table 1, we present the comparison of our model and SOTA methods on three EVC datasets with three different encoder settings, *i.e.*, ResNet-152, ResNet-101+3D-ResNext-101, and CLIP. We evaluate the performance of our model from three types of metrics. Our proposed model outperforms the state-of-the-art methods for all metrics, which demonstrates the superiority of our method.

**Table 2: The results of ablation studies on EVC-VE dataset to discuss on the effectiveness of our proposed components. A stands for our dynamic emotion perception; whereas, B represents the adaptive caption generation. "×" mark in B component means to leverage the decoder used in compared methods [1].**

| Components | | $Acc_{sw}$ | $Acc_c$ | BLEU-1 | BLEU-2 | BLEU-3 | BLEU-4 | METEOR | ROUGE | CIDEr | BFS | CFS |
| A | B | | | | | | | | | | | |
|---|---|---|---|---|---|---|---|---|---|---|---|---|
| × | × | 63.8 | 62.3 | 73.6 | 54.0 | 38.3 | 27.0 | 21.2 | 42.3 | 34.7 | 45.0 | 40.4 |
| ✓ | × | 68.7 | 67.2 | 73.2 | 54.5 | 39.1 | 27.4 | 21.8 | 46.7 | 38.7 | 46.4 | 44.9 |
| × | ✓ | 65.7 | 64.1 | 74.2 | 55.0 | 39.6 | 27.9 | 22.4 | 47.3 | 40.6 | 46.9 | 45.7 |
| ✓ | ✓ | **71.0** | **69.4** | **74.5** | **55.3** | **40.0** | **28.1** | **23.4** | **47.4** | **41.5** | **47.3** | **46.9** |

**Table 3: The results of ablation studies on EVC-VE dataset to discuss on the effectiveness of element-level emotion evolution and subspace-level emotion evolution, respectively.**

| Components | | $Acc_{sw}$ | $Acc_c$ | BLEU-1 | BLEU-2 | BLEU-3 | BLEU-4 | METEOR | ROUGE | CIDEr | BFS | CFS |
| Element | Subspace | | | | | | | | | | | |
|---|---|---|---|---|---|---|---|---|---|---|---|---|
| × | × | 63.8 | 62.3 | 73.6 | 54.0 | 38.3 | 27.0 | 21.2 | 42.3 | 34.7 | 45.0 | 40.4 |
| ✓ | × | 64.6 | 63.1 | 72.8 | 53.9 | 38.7 | 26.4 | 21.2 | 43.9 | 35.0 | 45.4 | 41.6 |
| × | ✓ | 67.4 | 66.5 | 73.1 | 54.3 | 38.9 | 27.1 | 21.5 | 45.0 | 37.3 | 45.9 | 43.1 |
| ✓ | ✓ | **68.7** | **67.2** | **73.2** | **54.5** | **39.1** | **27.4** | **21.8** | **46.7** | **38.7** | **46.4** | **44.9** |

**Table 4: Performance comparison for emotional image captioning task on SentiCap dataset.**

| Methods | $Acc_{sw}$ | $Acc_c$ | BLEU-1 | BLEU-2 | BLEU-3 | BLEU-4 | METEOR | ROUGE | CIDEr | BFS | CFS |
|---|---|---|---|---|---|---|---|---|---|---|---|
| SA-LSTM[39] | 83.4 | 83.4 | 43.3 | 24.7 | 14.0 | 7.9 | 13.9 | 34.2 | 34.8 | 30.0 | 44.5 |
| CANet[34] | 84.5 | 84.5 | 45.5 | 26.1 | 15.2 | 9.1 | 14.8 | 35.3 | 42.1 | 31.3 | 50.6 |
| EPAN[1] | 86.1 | 86.1 | 48.6 | 28.6 | 16.7 | 9.5 | 15.8 | 37.0 | 47.4 | 32.7 | 55.2 |
| **DCGN(Ours)** | **87.0** | **87.0** | **49.5** | **29.7** | **17.5** | **10.8** | **17.0** | **38.4** | **49.0** | **34.0** | **57.1** |

**Accuracy Evaluation.** Firstly, we evaluate the accuracy of our model in predicting video emotion categories. We observe that our method achieves the best performances on emotion metrics $Acc_{sw}$ and $Acc_c$ across the three datasets. For instance, on the EVC-VE dataset, the proposed method improves the performances by 4.8%/5.0% over EPAN[1] with ResNet-152 features on $Acc_{sw}$ and $Acc_c$. On EVC-MSVD and EVC-Combine datasets, the improvements are 2.8%/5.8% and 3.9%/2.9%, respectively. With ResNet-101+3D-ResNext-101 features, the improvements are more obvious, *i.e.*, 4.1%/5.6% and 5.1%/5.7% on EVC-VE and EVC-Combine dataset, since motion features can better capture changing emotional cues in videos. The remarkable improvements indicate that the emotion vectors can accurately capture changing emotions in videos.

**Semantic Evaluation.** Semantic metrics can reflect the quality of the captions generated by the model. We observe that our method achieves the best performance on multiple semantic metrics across the three datasets. It is worth noting that when using the modern CLIP features, our proposed method significantly outperforms the state-of-the-art methods. For instance, on the EVC-VE dataset, our model surpasses EPAN by a large margin of 9.8%, 15.7%, and 19.0% on BLEU-4, METEOR, and CIDEr, respectively. This substantial improvement demonstrates the effectiveness of leveraging CLIP features in generating more accurate, vivid, and diverse captions.

**Hybrid Evaluation.** In addition, we consider combining the emotion evaluation with BLEU and CIDEr metrics respectively to investigate the performance of our model. From the results in the last two columns of Table 1, our model achieves the best results on

the BFS and CFS metrics with modern CLIP features across EVC-VE, EVC-MSVD and EVC-Combine datasets. These show that our dual-path collaborative framework can mine the changing emotional cues implicit in the video and generate captions adaptively.

### 4.4 Ablation Studies

To further demonstrate the superiority of our proposed dual-path collaborative generation network, we conduct ablation studies for two perspectives discussion.

**Discussion on our proposed two path.** To further investigate the contribution of each path in our method, we conduct ablation studies, and the results are shown in Table 2. We first observe that the dynamic emotion perception could improve the effect by 4.9%, 4.9% on emotion prediction accuracy, which demonstrates that the dynamically changeable emotional cues in the video are precisely captured. However, its improvement on the BLEU metric is slight, and especially, there is even a decline on the BLEU-1 metric. A possible reason for these observations is that traditional caption generation framework incorrectly exploits dynamically changing emotion representations, leading to ambiguity in the generated descriptions. Thus, we propose the adaptive caption generation module in this work, which could dynamically estimate the intensity of emotion representations based on the alignment of the historical caption features and video features. Then, based on the estimated emotion intensity, we generate words adaptively by emotional guidance at different stages. To be specific, it can be seen that the adaptive caption generation path improves the

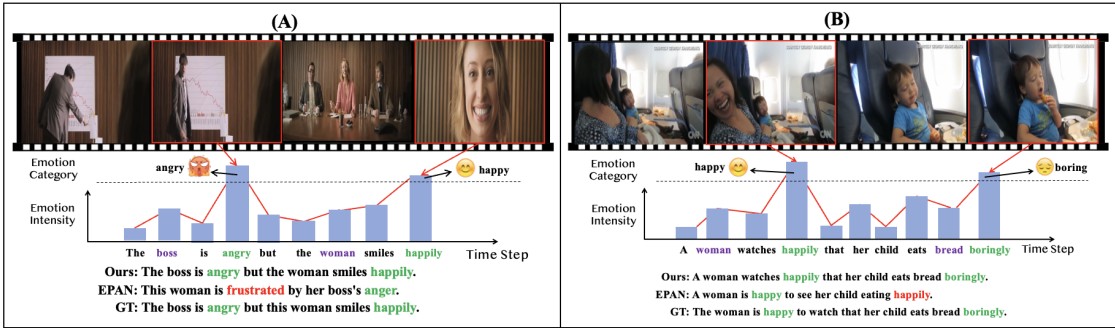

**Figure 5: Qualitative results on EVC-VE dataset. The figure shows different emotions at different stages of the videos. Moreover, we mark typical emotion-related and emotion-irrelevant words on our generated captions with green and purple respectively to demonstrate the effects of our model. The incorrect prediction is marked as red.**

performance by 0.9%, 1.3%, 1.7%, 1.1%, 5.1%, and 6.8% on semantic metrics and by 2.3% and 6.5% on hybrid metrics.

**Discussion on our emotion evolution methods.** In addition, we conduct ablation studies about the element-level and subspace-level evolution in dynamic emotion perception. The results are illustrated in Table 3. We firstly observe that the element-level evolution only brings a slight improvement, *i.e.*, 0.8, 0.8 on $Acc_{sw}$ and $Acc_c$, respectively. A possible reason is that the element-level evolution in our perception path serves as the global evolution without any strict restrictions, which means it avoid our methods falling into the local-optima. In contrast, the subspace-level emotion evolution explores multi-granular semantic information by reorganizing the dimensions of emotion features. Thus, it brings a remarkable improvements, *i.e.*, 3.6, 4.2 on $Acc_{sw}$ and $Acc_c$.

### 4.5 Qualitative Results

In order to qualitatively demonstrate the effect of our model, we make a visualization comparison with the state-of-the-art method EPAN on the EVC-VE dataset. Firstly, as shown in Figure 5, in case (A), EPAN incorrectly predicts the emotion word from "happily" to "frustrated". It is worth noting that "frustrated" emotion is quite opposite of the action "smile". Thus, since it incorporates the emotion into the caption generation for all time steps, the emotional cue "frustrated" is overemphsised, making it unable to perceive the "smile" action in the video and predict the factual word "smile". Instead, our method predicts the emotion evolution at each time step. Thus, our method successfully predicted the "smile" word in the caption generation by predicting the correct evolved emotions "happily". Secondly, in both two cases, since EPAN only perceive the global emotion, such as "frustrated" in case (A) and "happy" in case (B), their methods fail to predict the other emotion words in caption generation. For example, in case (B), the emotions changes to "boringly" as the video showing that the boy eat soullessly. However, EPAN generates wrong caption "child eating happily". Instead, our model correctly identified the emotional evolution from "happily" to "boringly" between the woman and the child in the video. Thus, we generates "a woman is happy to see" and "her child eats bread boringly" correctly with two different emotion guidance. Both two observations further demonstrate the effectiveness of our method.

### 4.6 Emotional Image Captioning

To further evaluate the generalization performance, we apply our method to the emotional image captioning task [12, 24, 40, 45, 46] on Senticap [30] dataset. Senticap dataset marks the positive/negative emotional polarity labels of the images based on the MS-COCO [27] dataset, and rewrites the description with emotions. We directly test the performance of our model on the Senticap dataset. The results are shown in Table 4. It can be seen that our method achieves the best performances on all metrics, *i.e.*, 3.8%, 7.8%, 4.0% on BLEU-2, METEOR, and BFS, respectively. Although compared with videos, images lose the temporal information and are less likely to have dynamic changes in emotion, our dynamic emotion perception module can still contribute to emotional bias correction. Even if the initial emotion perception is wrong, our dynamic evolution module help to correct the emotional bias during the caption generation. In the step where emotion-related words need to be generated, the correct emotion can be perceived and guide the word generation.

### 5 CONCLUSION

In this paper, we propose a novel Dual-path Collaborative Generation Network for emotional video captioning, which dynamically perceives visual emotional cues evolutions while generating emotional captions by collaborative learning. The two paths promote each other and significantly improve the emotional caption generation performance. To be specific, a dynamic emotion evolution module, including element-level emotion evolution and subspace-level emotion evolution, is designed in the dynamic emotion perception path, which dynamically recomposes emotion features to achieve emotion evolution at different generation steps, helping to correct the previous emotion bias and provide customized and accurate emotion guidance for each generation step. Besides, an emotion adaptive decoder is proposed in the adaptive caption generation path that estimates the emotion intensity for each step, helps to generate emotion-related words at the necessary time step, and balances the guidance role of the factual content and emotional cues. Extensive experiments on three challenging datasets demonstrate the superiority of our method and each proposed module.

# 6 ACKNOWLEDGMENTS

This work is supported by the National Natural Science Foundation of China under Grant 62302474, and the National Science Fund for Excellent Young Scholars under Grant 62222212.

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
