# OpenReview forum: "Dual-path Collaborative Generation Network for Emotional Video Captioning"
_acmmm.org/ACMMM/2024/Conference — MM2024 Oral_

### Official Review · Reviewer_ZRmv · 2024-05-20

**Rating:** 6
**Confidence:** 4

**Summary:**

In this paper, the authors propose a novel Dual-path Collaborative Generation Network for emotional video captioning, which dynamically perceives visual emotional cues evolutions while generating emotional captions by collaborative learning. The two paths promote each other and significantly improve the emotional caption generation performance. Specifically, a dynamic emotion evolution module, including element- and subspace-level evolution, is designed in the dynamic emotion perception path, which dynamically recomposes emotion features to achieve emotion evolution at different generation steps, helping to correct the previous emotion bias and providing customized and accurate emotion guidance for each generation step. Besides, an emotion adaptive decoder is proposed in the adaptive caption generation path that estimates the emotion intensity for each step, helps to generate emotion-related words at the necessary time step, and balances the guidance role of the factual content and emotional cues. Extensive experiments on three challenging datasets demonstrate the superiority of the proposed method and each proposed module.

**Strengths:**

- Novelty: This paper introduces a novel collaborative learning framework to integrate dynamic emotion perception into the emotional video caption generation. Moreover, an innovative dynamic emotion evolution module is proposed, which helps to obtain fine-grained emotion representation by two cascade evolution modules including element- and subspace-level evolution, thereby providing more accurate emotion guidance for caption generation at different time steps.
- Quality: This paper has an exhaustive comparison with SoTA using multiple natural language generation metrics and emotion recognition accuracy metrics. The proposed methods showed significant gains on all three prevailing EVC datasets. The authors also present several qualitative results to investigate how the proposed approach perceives accurate emotions for each generation step.
- Clarity: This paper is well-structured and easy to follow.
- Significance: Leveraging collaborative learning to perceive the subtle emotional cues to boost emotional video captioning is important for multimedia communities. The emotion-based multimedia understanding is a hot topic today and yet under-researched.

**Limitations:**

1. Please quantify the experimental results in the Abstract.
2. Please revise your qualitative results. The existing results can only show that the proposed method works better in the case of multiple emotions, but does not illustrate the superiority of the single emotion case.
3. Some occasional writing typos need to be revised:
    - In line 382, “this” should be “these”.
    - In line 441, “mechanisms” should be “mechanism”.

**Suitability:**

3

---

### Official Review · Reviewer_FeWS · 2024-05-24

**Rating:** 4
**Confidence:** 3

**Summary:**

This paper proposes a novel method for emotional video captioning (EVC). The motivation of this paper is to deal with two challenges in EVC: (1) It is required to capture subtle changes of emotions during the video, which results in diverse  captions. (2) It is required to assign different weights to emotions and factual contents in different steps to avoid overemphasizing emotions when generating captions. To solve these problems, this paper designs a dual-path collaborative generation network, which includes a dynamic emotion module to achieve emotion evolution at different generation steps and an emotion adaptive decoder to balance the factual content and emotional cues.

**Strengths:**

1. This paper proposes a novel method to deal with emotional video captioning (EVC). The proposed method involves two modules, one of which is a dynamic emotion module to achieve emotion evolution at different generation steps, and the other of which is an emotion adaptive decoder to balance the factual content and emotional cues.
2. The proposed method is superior to current SOTAs on EVC with a remarkable improvement.
3. The representation of this paper is clear.

**Limitations:**

1. As far as I know, there have existed many advanced video captioning methods these years, like some cross modal transformers[1]. But this paper only compares with one video captioning method which was proposed in 2018. It is possible to question the value of this paper if current advanced video captioning works can perform well on EVC.
2. For experiment of emotional image captioning on SentiCap dataset, there lacks  emotional image captioning methods like[2,3] which makes the conclusion unconvincing.
3. It is better to include some details in this paper for reader understanding. For example, the calculation metric of BFS and CFS, which is not common in video captioning.

[1] VideoBERT: A Joint Model for Video and Language Representation Learning, CVPR, 2019

[2] Senticap: Generating image descriptions with sentiments, AAAI, 2016

[3] Visual Captioning at Will: Describing Images and Videos Guided by a Few Stylized Sentences, MM, 2023

**Suitability:**

3

---

### Official Review · Reviewer_3Ecw · 2024-05-24

**Rating:** 3
**Confidence:** 3

**Summary:**

This work proposes a dual-path collaborative generation network for emotional video captioning. It combines dynamic emotion perception and adaptive caption generation, which can effectively perceive emotional changes in videos, and balance emotion and factual content in subtitle generation, so as to realize the optimization of emotional video subtitle generation.

**Strengths:**

1. This paper proposes a dual-path collaborative generation network, which combines dynamic emotion perception and adaptive caption generation to achieve more accurate, vivid and diversified subtitle generation, which is innovative.

2. The experiment is sufficient. The author not only uses multiple data sets to prove the performance of the proposed model, but also uses different types of ablation experiments and a variety of indicators to prove the advantages of the proposed module

**Limitations:**

1. A large number of experimental results have been obtained in the paper. In the "Semantic Evaluation" part of section 4.3 of the experiment, the performance improvement data described by the author may be wrong. It is suggested that the author re-check the calculation accuracy of each data in the paper.

2. In the "Discussion on our proposed two path" part of section 4.4 of the experimental part, the author's last sentence of performance improvement did not clearly indicate the corresponding specific indicators, which is easy to confuse readers. It is suggested that the author point out the corresponding indicators in detail.

3. The author used "EmVidCap-S, EmVidCap-L, EmVidCap" when introducing the data set in Section 4.1 of the experiment, but the data set name in Table 1 is "EVC-MSVD, EVC-VE, EVC-Combine", which is easy to be misunderstood by readers outside the neighborhood.

**Suitability:**

2

---

### Official Review · Reviewer_EN7N · 2024-05-25

**Rating:** 5
**Confidence:** 2

**Summary:**

This work proposes a new method for the Emotion Video Captioning (EVC) task called Dual-path Collaborative Generation Network (DCGN). The authors emphasize capturing the emotional shifts within the video for better emotion-captioning. To do so, DCGN consists of two paths- A dynamic Emotion path, which captures the subtle emotional evolution considering the old captioning and visual context, and an Adaptive Captioning path, which generates the caption considering the evolved emotion cue from the dynamic emotion path. Details of both the branches are well explained. DCGN is shown to be superior over prior works on three public video EVC datasets and has also proven effective for Emotional Image Captioning (EIC) tasks. Sufficient experiments, and ablation studies are performed to justify the design choices.

**Strengths:**

1. All the modules of DGCN are well explained, and the intuition behind each module is included.
2. Authors conducted appropriate ablation studies to justify the design choices.
3. Qualitative result reflects the importance of a dynamic emotion path, which is able to correctly the characters and their emotions.

**Limitations:**

1. Considering the abilities of recent Large Vision Language Models (LVLMs) or Large Multi-Modal Models (LMMMs) like GPT-4o, I encourage authors to include such models as a baseline as well.
2. The paper is too wordy and not easy to read. E.g. abstract lines 27 to 34 are one continuous line that is tough to follow. I encourage authors to simplify the language wherever possible. Many claims/explanations made in the Introduction section get cleared when the reader reaches the dedicated section.
3. In section 4.2, please also comment on the resources required to use DCGN.
4. Authors have used ACMM 2017 template. Please transfer the contents to 2024 template.

**Suitability:**

3

---

### Meta-Review · Area_Chair_JQbf · 2024-06-25

**Recommendation:** Accept (Oral)
**Confidence:** 5

**Metareview:**

All reviewers are satisfied with the responses and the contributions in the filed, I recommend its acceptance.